# Arrhenius Crossover Temperature of Glass-Forming Liquids Predicted by an Artificial Neural Network

**DOI:** 10.3390/ma16031127

**Published:** 2023-01-28

**Authors:** Bulat N. Galimzyanov, Maria A. Doronina, Anatolii V. Mokshin

**Affiliations:** 1Institute of Physics, Kazan Federal University, 420008 Kazan, Russia; 2Udmurt Federal Research Center of the Ural Branch of RAS, 426067 Izhevsk, Russia

**Keywords:** machine learning, physical properties, organic compounds, metallic alloys, silicates, borates

## Abstract

The Arrhenius crossover temperature, $T_{A}$, corresponds to a thermodynamic state wherein the atomistic dynamics of a liquid becomes heterogeneous and cooperative; and the activation barrier of diffusion dynamics becomes temperature-dependent at temperatures below $T_{A}$. The theoretical estimation of this temperature is difficult for some types of materials, especially silicates and borates. In these materials, self-diffusion as a function of the temperature *T* is reproduced by the Arrhenius law, where the activation barrier practically independent on the temperature *T*. The purpose of the present work was to establish the relationship between the Arrhenius crossover temperature $T_{A}$ and the physical properties of liquids directly related to their glass-forming ability. Using a machine learning model, the crossover temperature $T_{A}$ was calculated for silicates, borates, organic compounds and metal melts of various compositions. The empirical values of the glass transition temperature $T_{g}$, the melting temperature $T_{m}$, the ratio of these temperatures $T_{g}/T_{m}$ and the fragility index *m* were applied as input parameters. It has been established that the temperatures $T_{g}$ and $T_{m}$ are significant parameters, whereas their ratio $T_{g}/T_{m}$ and the fragility index *m* do not correlate much with the temperature $T_{A}$. An important result of the present work is the analytical equation relating the temperatures $T_{g}$, $T_{m}$ and $T_{A}$, and that, from the algebraic point of view, is the equation for a second-order curved surface. It was shown that this equation allows one to correctly estimate the temperature $T_{A}$ for a large class of materials, regardless of their compositions and glass-forming abilities.

## 1. Introduction

In the last decade, interest in the study of phase transformations in glass-forming liquids has increased significantly [1,2,3]. There is increasing evidence that such transformations can be related to the ability of a liquid to form a glassy state [4,5,6]. The results of recent studies show that the glass-forming ability of a liquid depends on the specifics of changes in its atomistic structure and collective dynamics near the melting temperature $T_{m}$ [7,8,9]. The beginning of such the changes in the dynamics of a liquid corresponds to the Arrhenius crossover temperature $T_{A}$ [10,11,12,13]. It is generally accepted that the atoms of a liquid do not form any bound structures above $T_{A}$. In this case, the dependence of the logarithm of viscosity on the reverse temperature obeys a linear law (so-called high-temperature Arrhenius behavior). Below $T_{A}$, individual groups of atoms become less mobile, which manifests in the deviation of viscosity from the Arrhenius behavior, which is typical for equilibrium liquids [14,15,16].

The existing empirical and theoretical methods for estimating $T_{A}$ are mainly based on analysis of the temperature-dependence of liquid viscosity (or the structural relaxation time) and on determining the high-temperature linear regime in this relationship [17,18,19,20]. As a rule, linear approximation methods most accurately characterize this linear regime. Such approximations are applicable only if the viscosity of the liquid is determined for a wide temperature range covering temperatures above and below the melting temperature ($T_{m}$). For organic (molecular) compounds and polymers belonging to the class of the so-called *fragile glass formers*, viscosity increases rapidly with decreasing temperature, which makes it possible to determine the deviation from the high-temperature Arrhenius behavior. For the so-called *strong glass formers*, including most metal melts, silicates and borates, the Arrhenius behavior practically does not change even when passing through the melting temperature and entering the region of supercooled melt. This is displayed in blurring the region of transition from the high-temperature Arrhenius behavior to the low-temperature nonlinear regime. Therefore, the accuracy of the temperature estimation can be low, and the estimated values of $T_{A}$ practically do not correlate with the other physical characteristics of liquids. For example, an expression was proposed by A. Jaiswal et al. which relates the fragility index *m* with the $T_{A}$ values of various glass-forming liquids. This expression takes into account the temperature dependence of the transport properties (mainly self-diffusion) and the dynamics of atoms near the glass transition [8]. This expression gives a correct correspondence between *m* and $T_{A}$ in the case of molecular glasses, though the results of calculations can differ greatly from empirical data in the case of metallic and optical glasses. Further, the analytical expression was proposed by T. Wen at al., according to which the glass-forming ability of liquid is related to the reverse temperature $1/T_{A}$: i.e., the higher the $T_{A}$, the worse the liquid forms a stable glassy state [21]. However, this rule is valid only for a narrow class of glass formers that are similar in composition (mainly for metallic glasses). Therefore, obtaining an analytical expression that allows one to determine $T_{A}$ based on the known key physical characteristics of glass-forming liquids remains an unsolved task. It is obvious that the correct solution of this task is possible using machine learning methods, which will allow us to reveal hidden relationships between physical characteristics and determine the most significant factors in estimating $T_{A}$ [22,23,24,25,26].

The purpose of the present study was to determine how physical characteristics associated with the *overall kinetics* of supercooled liquids correlate with each other. These characteristics are primarily

(i)the glass transition temperature ($T_{g}$), at which liquid becomes amorphous upon rapid cooling,(ii)the melting temperature ($T_{m}$),(iii)the Arrhenius crossover temperature ($T_{A}$),(iv)the Kauzmann temperature,(v)the high-temperature limit ($T_{\infty }$), at which the viscosity tends to zero,(vi)the temperature ($T_{0}$) associated with the transition to a non-ergodic phase (for example, in the mode-coupling theory),(vii)the temperature ratio of $T_{g}/T_{m}$, which is considered as one of the criteria for the glass-forming ability of liquids and(viii)the fragility index *m*, which determines the rate of change in viscosity with temperature.

Some of these characteristics come to the fore for several reasons. First of all, these characteristics are available for experimental measurements. In addition, they are presented in various models that reproduce the kinetics and transport properties of supercooled melts. Model equations for the shear viscosity—such as the equations of the Vogel–Fulcher–Tammann–Hesse [12], Mauro et al. [27], Avramov–Milchev [28] and the equation obtained in the framework of the mode-coupling theory [29]—contain three or even more parameters to reproduce the viscosity over a range of the temperatures. This indicates that it is necessary to consider some temperature pairs associated with the supercooled melt phase. It is important to note that these temperature pairs occur in arbitrary combinations, which indirectly indicates the presence of correlations between “critical” temperatures in some way related to the glass transition. Moreover, this fact is directly supported by previous results relating to the description of the temperature dependence of the viscosity and crystallization rate characteristics of supercooled melts by the scale relations [3,9,30], where only the melting and glass transition temperatures, $T_{m}$ and $T_{g}$, appear as input parameters. Thus, the determination of specific correlation relationships between the “critical” temperatures of the kinetics of viscous melts is an important task, the solution of which will contribute to a deeper understanding of the solidification processes (glass transition and crystallization).

In the present work, the Arrhenius crossover temperature $T_{A}$ is predicted for various types of glass-forming liquids, including silicates, borates, metal melts and organic compounds using the machine learning method. The most significant factors among the physical characteristics of these glass-forming liquids are determined. Taking into account these factors, an analytical equation is obtained that allows one to accurately relate the temperature $T_{A}$ to the physical properties of glass-forming liquids.

## 2. Data Set and Machine Learning Model

Using an appropriate set of physical properties as the neural network input parameters is a crucial for correct predicting the Arrhenius crossover temperature. These physical properties must uniquely characterize the nature of the material and must be determined with high accuracy by experimental or simulation methods. Here, it is quite reasonable to choose the fragility index (*m*), the melting temperature ($T_{m}$), the glass transition temperature ($T_{g}$) and the so-called reduced glass transition temperature ($T_{g}/T_{m}$), whose values are known for almost all types of glass-forming liquids and can be found in the scientific literature. Moreover, for some organic and metallic glass formers, the phenomenological relation between $T_{g}$ and $T_{A}$ is known [5,15]. For most silicates and borates, there is no known correlation between these two temperatures. At the same time, there can be hidden relationships, which are usually revealed using machine learning methods.

The initial data set for machine learning included experimental and calculated data as well as information from databases (e.g., ITPhyMS-Information technologies in physical materials science, Materials Project) [8,12,31]. For our purpose, different glass-forming materials were selected, among which were silicates, borates, organic compounds and metallic alloys (Cu, Zr, Ti, Ni, Pd-based) (see Appendix A). We chose systems for which the melting temperature, the glass transition temperature and the fragility index are known. This data set was divided into the sets corresponding to *training*, *validation* and *test* regimes. The *training* and *validation* sets included all organic compounds and metallic alloys, along with several silicates and borates, for which $T_{A}$ is known. The machine learning model was created on the basis of the training data set. The accuracy of the neural network was checked using the validation data set. The *test* set included only silicates and borates, for which $T_{A}$ was predicted. Note that to create an artificial neural network, we used instances for which all parameters are known. Predictions were made only for those systems for which the temperature $T_{A}$ is unknown. The reliability of the obtained results is quite expected, since the formation of the neural network was performed using the data for systems of all categories, including those for which further predictions were made.

In the present work, the machine learning model was a feedforward artificial neural network (see Figure 1). This model has one input layer with four neurons, for which the values of the melting temperature, the glass transition temperature, the ratio $T_{g}/T_{m}$ and the fragility index were taken from the data set. The values of this physical characteristics were renormalized and presented in the range [0, 1]. Next two layers of the neural network were hidden and consisted of 20 neurons. In the output layer we had only one neuron, which determined $T_{A}$. For the initiation of the neural network, the values of all neurons and their weight coefficients were assigned randomly from the range [0, 1]. Subsequently, calculation of the values of all neurons was carried out as follows [32]:(1)$$n_{i}^{\left(k\right)}=f\left(\sum_{j=1}^{N_{k-1}}w_{ij}^{\left(k-1\right)}n_{j}^{\left(k-1\right)}+b_{i}^{\left(k\right)}\right).$$

Here, $n_{i}^{\left(k\right)}$ is the value of the *i*th neuron in the *k*th layer (k=2,3,4); $w_{ij}^{\left(k-1\right)}$ is the value of the (k−1)th layer weight going from a neuron with index *j* to a neuron with index *i* from the *k*th layer; $b_{i}^{\left(k\right)}$ is the bias weight acting on a neuron with index *i* from the *k*th layer; $N_{k-1}$ is the number of neurons in the (k−1)th layer; function f(…) is the sigmoid-type logistic function:(2)$$f\left(x\right)=\frac{1}{1-\exp(-x)}.$$

The neural network was trained using the backpropagation algorithm, according to which the value of the weight coefficient was adjusted as follows [33]:(3)$$w_{ij}^{(k),\;new}=w_{ij}^{\left(k\right)}-\gamma \frac{\partial \chi (s)}{\partial w_{ij}^{\left(k\right)}}.$$

γ is the training rate, the value of which is usually chosen in the range [0, 1]. In the present work, we took the rate of γ=0.3 as optimal for the considered machine learning model. The value of the loss function χ(s) is determined as
(4)$$\chi \left(s\right)=\frac{1}{2}{ [n_{1}^{\left(4\right)}\left(s,l\right)-n\left(l\right)]}^{2},$$
where *s* is the training iteration number (i.e., epoch number); $n_{1}^{\left(4\right)}\left(s,l\right)$ is the value of the output neuron at the *s*th epoch for the *l*th element from the training data set; n(l) is the required value of the output neuron for the *l*th element. To train the machine learning model, 2400 epochs were used. The gradient of the loss function with respect to each weight was computed by the chain rule, according to which Equation (Equation 3) can be represented in the following form:(5)$$w_{ij}^{(k),\;new}=w_{ij}^{\left(k\right)}-\gamma \delta_{i}n_{i}^{\left(k\right)}\frac{e^{-W_{i}^{\left(k\right)}}}{{ [1+e^{-W_{i}^{\left(k\right)}}]}^{2}},$$
where
$$\delta_{i}=\left\{\begin{array}{cc} n_{1}^{\left(4\right)}\left(l\right)-n\left(l\right) & \mathrm{if}\;i\;\mathrm{is}\;\mathrm{the}\;\mathrm{output}\;\mathrm{layer}\;\mathrm{neuron}\\ \sum_{j}w_{ij}\delta_{j} & \mathrm{if}\;i\;\mathrm{is}\;a\;\mathrm{neuron}\;\mathrm{of}\;\mathrm{the}\;\mathrm{hidden}\;\mathrm{layers} \end{array}\right.,$$
(6)$$W_{i}^{\left(k\right)}=\sum_{j=1}^{N_{k-1}}w_{ij}^{\left(k-1\right)}n_{j}^{\left(k-1\right)}.$$

This backpropagation algorithm allows one to control the training procedure. The criterion for finishing this procedure is the minimal error between the results of the output neuron and the required values from the validation data set.

## 3. Identification of Significant Physical Properties

To identify the physical characteristics that are most significant for estimating the temperature $T_{A}$, calculations were carried out for various combinations of the neural network’s input parameters. As shown in Figure 2a, retraining of the machine learning model was performed for various combinations of $T_{m}$, $T_{g}$, $T_{g}/T_{m}$ and *m* using the training and validation data sets. For each considered combination, the root mean square error was calculated:(7)$$\xi =\sqrt{\frac{1}{N}\sum_{i=1}^{N}\left(T_{A}^{\left(pred\right)}-T_{A}^{\left(emp\right)}\right)}.$$

Obviously, the smaller the value of ξ, the more accurately $T_{A}$ is determined. In Equation (Equation 7), $T_{A}^{\left(pred\right)}$ is the Arrhenius crossover temperature predicted by machine learning model; $T_{A}^{\left(emp\right)}$ is the empirical Arrhenius crossover temperature; *N* is the number of elements in the data set. The obtained results reveal that the most significant physical quantities correlating with $T_{A}$ are the glass transition temperature $T_{g}$ and the melting temperature $T_{m}$. This is confirmed by the relatively small value of the mean absolute error, which does not exceed ξ=11.4 K. The quantities $T_{g}/T_{m}$ and *m* are less significant in the estimation of the temperature $T_{A}$, which is clearly manifested in the relatively large ξ with values of up to 25.8 K. The smallest error ξ≈10.5 K is obtained by taking into account all four physical quantities at which the best agreement between the empirical values of $T_{A}$ and the result of the machine learning model is observed.

Figure 2b shows that the empirical and predicted temperatures $T_{A}$ correlate well with each other. Regarding organic compounds, an insignificant variation between the empirical and predicted $T_{A}$ can be observed for saccharides (for example, fructose, trehalose). This was mainly due to insufficient of data in the training set for this class of materials. For metal melts, the variation in the values of $T_{A}$ can be observed for alloys based on rare earth elements (for example, Pr${}_{60}$Ni${}_{10}$Cu${}_{20}$Al${}_{10}$). The empirical and predicted values of $T_{A}$ have a minimum variation for silicates and borates. This result indicates that artificial neural networks have good trainability with respect to these materials. The reason for this could be that the change in viscosity of a silicate and borate melt occurs in a similar way in a wide temperature range, including near the melting temperature [12]. Such universality in the temperature dependencies of viscosity is kept when the composition of the melts changes, for example, by adding alkali oxides (${\mathrm{Li}}_{2}$O, ${\mathrm{Na}}_{2}$O, $K_{2}$O, etc.) or metal oxides (${\mathrm{Al}}_{2}O_{3}$, MgO, PbO, etc.).

## 4. Regression Model for Arrhenius Crossover Temperature

Figure 3a shows the correspondence between the glass transition temperature $T_{g}$ and the predicted temperature $T_{A}$ for various glass-forming liquids. For organic compounds, the correspondence between $T_{A}$ and $T_{g}$ is reproduced according to the linear law $T_{A}≃k\cdot T_{g}$ with k=1.4. It is noteworthy that this correspondence between $T_{A}$ and $T_{g}$ was predicted earlier (for example, see Refs. [8,34]). For metallic glass formers, there is a relationship between $T_{A}$ and $T_{g}$ of the form $T_{A}=k\cdot T_{g}$, where k=2.0±0.2. As a rule, such a relationship between temperatures $T_{A}$ and $T_{g}$ is universal for metal alloys containing two to five different components [8]. For silicates and borates, there is no clear correlation between $T_{A}$ and $T_{g}$: the known laws do not reproduce the correspondence between $T_{A}$ and $T_{g}$. The results given in Figure 3b reveal the obvious correlation between $T_{A}$ and $T_{m}$ for silicates and borates, whereas variation in values of these temperatures is more pronounced for organic and metallic glass formers. Despite this, the correspondence between $T_{A}$ and $T_{m}$ is reproduced by the linear law regardless of the type of glass-forming liquid.
(8)$$T_{A}=k\cdot T_{m}\;\;\;\left(\mathrm{where}\;\;k=1.1\pm 0.15\right)$$
It is noteworthy that this result agrees with the results of Refs. [35,36].

Relationship (Equation 8) is an empirical result that has no theoretical explanation and is only *an approximation*. The error of this relationship depends both on the specific type of material and on the category to which this material belongs (i.e., organic, metallic, silicate). As shown in Figure 3b, relationship (Equation 8) only qualitatively reproduces the empirical data for a large data set. At the same time, one can be convinced that for certain categories of materials, this relationship yields very accurate results. Thus, for example, the available data for organic materials and metallic systems are more correctly reproduced by the quadratic polynomials than by the linear relationship (see Figure 3b). On the other hand, the results for silicates and borates reveal a general trend of increasing $T_{A}$ with $T_{m}$, which can be described by the linear relationship $T_{A}=aT_{m}+b$, where the parameters *a* and *b* take different values for materials from different categories. In this regard, it is quite natural to expect that the *overall correlation* between $T_{A}$ and $T_{m}$ is not as so simple as prescribed by relationship (Equation 8), and it requires taking into account other physical characteristics.

For *implicit* (“hidden”) correlations between different parameters, it is quite natural and often feasible that the parameters do not appear in the resulting correlation relation as single additive terms, but in the form of combinations (products or ratios). For example, in contrast to the methodology of artificial neural networks, this is most clearly manifested in the method of joint accounting for arguments using the Kolmogorov–Gabor polynomial [37,38,39]:(9)$$y\left(x_{1},\ldots ,x_{n}\right)=a_{0}+\sum_{i=1}^{n}a_{i}x_{i}+\sum_{i=1}^{n}\sum_{j=i}^{n}a_{ij}x_{i}x_{j}+\sum_{i=1}^{n}\sum_{j=i}^{n}\sum_{k=j=1}^{n}a_{ijk}x_{i}x_{j}x_{k}+\ldots ,$$
which determines the relationship of a parameter *y* with the parameters $x_{1}$, $x_{2}$,… $x_{i}$,… In the obtained model of the artificial neural network, the appearance of the parameter $T_{g}/T_{m}$, together with the individual parameters $T_{g}$ and $T_{m}$, directly indicates that the Arrhenius crossover temperature $T_{A}$ correlates not only with the absolute values of the melting and glass transition temperatures for different systems, but also with their ratio. This result is fully consistent with the theoretical description of crystallization rate characteristics of supercooled melts within the reduced temperature scale ${\tilde{T}}_{MG}$ and universal scaled relations [9,30]. This point is discussed in detail in Ref. [30] (see text on page 104502-2).

To obtain a general expression relating the temperatures $T_{g}$, $T_{m}$ and $T_{A}$, the reproducibility of these temperatures was tested in the framework of the nonlinear regression model:(10)$$T_{A}\left(T_{g},T_{m}\right)=\sum_{i=1}^{k}\left(a_{i}T_{g}^{i}+b_{i}T_{m}^{i}+c_{i}T_{g}^{i}T_{m}^{i}\right).$$

The temperatures $T_{g}$ and $T_{m}$ are input parameters determined by experimental values; the temperature $T_{A}$ is the resulting factor; *k* is an integer value chosen during the regression analysis; $a_{i}$, $b_{i}$ and $c_{i}$ are the fitting coefficients, whose values are determined by enumeration to obtain the best agreement between the empirical temperature $T_{A}$ and the result of Equation (Equation 10).

The values of the fitting coefficients were determined by regression analysis: $a_{1}=b_{1}=0.7016$, $a_{2}=-7.52\times 10^{-4}$ K${}^{-1}$, $c_{1}=4.43\times 10^{-4}$ K${}^{-1}$. As was found, all other fitting coefficients equal zero. Thus, with these values of the fitting coefficients, we obtained the minimum error between the empirical $T_{A}$ and the result of Equation (Equation 10) for the considered glass-forming liquids. Thus, the temperatures $T_{g}$, $T_{m}$ and $T_{A}$ can be related by the nonlinear regression model:(11)$$T_{A}\left(T_{g},T_{m}\right)=a_{1}T_{g}+a_{2}T_{g}^{2}+b_{1}T_{m}+c_{1}T_{g}T_{m}.$$

In algebra, an equation of this type is known as the equation of a second-order curved surface. Figure 4 shows that Equation (Equation 11) correctly determines the correspondence between the temperatures $T_{g}$, $T_{m}$ and $T_{A}$ for all considered glass formers. The average error between the empirical data and the result of Equation (Equation 11) is ∼10%. The plane surface corresponds to the data for organic compounds and metal melts. The deviation from this surface and its transformation into a curved surface occurs due to taking into account the data for silicates and borates (see Figure 4b). Therefore, Equation (Equation 11) can be applied to determine $T_{A}$ for various types of materials, regardless of composition.

Note that Equation (Equation 11) is an empirical result, the rigorous physical meaning of which has not yet been established. This is also true for relationship (Equation 8), which also has no a clear physical meaning. On the other hand, Equation (Equation 11) shows that the three key temperatures associated with a change in kinetic regime (as in the case of $T_{A}$) and with a change in thermodynamic phase (as for $T_{m}$ and $T_{g}$) correlate in some universal way with each other for melts that are different in physical nature. The necessity of the quadratic contribution in Equation (Equation 11) to reproduce the empirical data becomes obvious if these data are represented in the space of three parameters—temperatures $T_{A}$, $T_{m}$ and $T_{g}$—as shown in Figure 4b. As can be clearly seen in this representation, the empirical data form the second-order curved surface, for the analytical reproduction of which the presence of the quadratic contributions $T_{g}^{2}$ and $T_{g}T_{m}$, are necessary. Moreover, since the curvature of this surface is expressed significantly, its projection onto the coordinate plane ($T_{A}$, $T_{m}$) will give a certain curve that can be reproduced by a straight line only *approximately* (for example, as prescribed by relationship (Equation 8)). It should be noted that such representation of the empirical data in ($T_{A}$, $T_{m}$, $T_{g}$)-space was not expected and originally carried out; and Equation (Equation 11) is a direct result of the regression analysis.

To determine the error in estimating $T_{A}$ for materials of various categories, the root mean square relative error (RMSRE) was calculated:(12)$$\mathrm{RMSRE}=\sqrt{\frac{1}{n}\sum_{i=1}^{n}{\left(\frac{T_{A}^{\left(emp\right)}-T_{A}^{\left(calc\right)}}{T_{A}^{\left(emp\right)}}\right)}^{2}},$$
where $T_{A}^{\left(emp\right)}$ is the empirical value of $T_{A}$; $T_{A}^{\left(calc\right)}$ is the $T_{A}$ computed by various methods—a machine learning model, by relationship (Equation 8) and by Equation (Equation 11). Figure 5 shows that the accuracy at estimation of $T_{A}$ depends on the applied method and the category of material. Thus, for silicates and borates, the error of machine learning prediction is lower than that of other methods. In this case, Equation (Equation 11) is more accurate than relationship (Equation 8). For metallic systems, the errors of all methods are comparable, although Equation (Equation 11) produces the smallest error. For organic materials, the machine learning prediction is more accurate than other methods. In this case, the error of Equation (Equation 11) is higher than the error of relationship (Equation 8). This is due to the fact that for materials with complex structures, such as organic materials, the glass transition temperature is determined ambiguously. Namely, for this category of materials, the temperature $T_{g}$ in relation to the melting temperature $T_{m}$ can vary widely compared to silicate, borate and metallic systems. For example, for organic materials, the variation in $T_{g}/T_{m}$ exceeds 0.5, whereas for borate, silicate and metallic systems this variation is usually less than 0.4.

## 5. Conclusions

The physical characteristics of various type glass-forming liquids were determined using a machine learning model—that is, those which are most significant to correct prediction/estimation of the Arrhenius crossover temperature. Such significant factors are the glass transition temperature and the melting temperature. It has been established that the fragility index and the reduced glass transition temperature ($T_{g}/T_{g}$), which is directly related to the glass-forming ability of a liquid, are insignificant factors. These factors do not affect the accuracy of $T_{A}$ estimation. The correctness of the obtained results was confirmed by the presence of a good correlation between the empirical values of $T_{A}$ and the $T_{A}$ predicted by a machine learning model. Moreover, the result of the machine learning model gives the correct relationships between the temperatures $T_{A}$, $T_{g}$ and $T_{m}$, which agree with the previously established empirical rules $T_{A}≃1.1T_{m}$ (for all types of liquids), $T_{A}≃1.4T_{g}$ (for organic compounds) and $T_{A}≃2.0T_{g}$ (for metallic systems). Based on the results of nonlinear regression analysis, an equation was obtained that allows one to determine the temperature $T_{A}$ by using known temperatures $T_{g}$ and $T_{m}$. It was shown that this equation gives the correct values of $T_{A}$ for various types of liquids, including silicates and borates, for which direct estimation of $T_{A}$ can be difficult.

## Figures and Tables

**Figure 1 materials-16-01127-f001:**
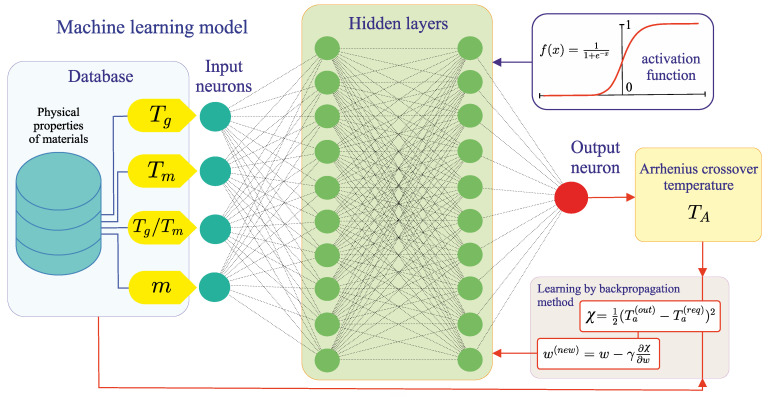
Scheme of the machine learning model based on the feedforward artificial neural network.

**Figure 2 materials-16-01127-f002:**
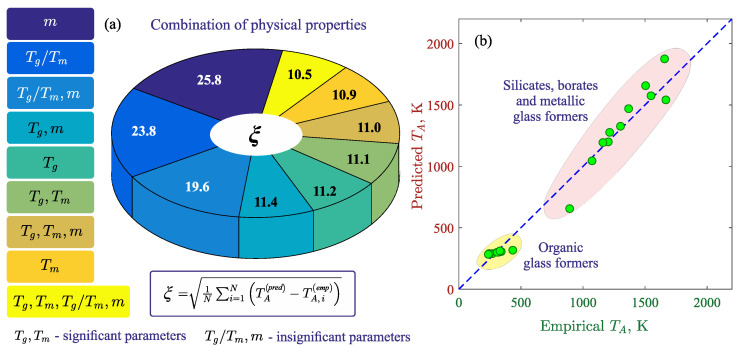
(**a**) Diagram of the root mean square error ξ of estimation of the Arrhenius crossover temperature $T_{A}$ calculated for various combinations of the quantities $T_{m}$, $T_{g}$, $T_{g}/T_{m}$ and *m*, which were the inputs of the machine learning model. Inset: $T_{A}^{\left(pred\right)}$ and $T_{A}^{\left(emp\right)}$ are the predicted and empirical Arrhenius crossover temperatures, respectively. (**b**) Correspondence between the empirical $T_{A}$ and the $T_{A}$ predicted by the machine learning model using the validation data set.

**Figure 3 materials-16-01127-f003:**
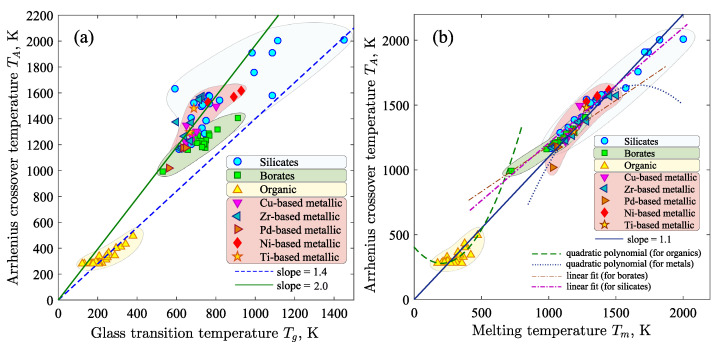
(**a**) Correspondence between the glass transition temperature $T_{g}$ and the predicted value of the Arrhenius crossover temperature $T_{A}$ for different types of glass formers. (**b**) Correlation between the melting temperature $T_{m}$ and the predicted temperature $T_{A}$. The dashed and dotted lines show the interpolation by the quadratic polynomials: $T_{A}=409-1.23T_{m}+0.003T_{m}^{2}$ in the case of organic materials and $T_{A}=-2161+4.6T_{m}-0.0014T_{m}^{2}$ for metallic systems. The dot-dash lines show the linear fit by equations $T_{A}=318+0.9T_{m}$ (for silicates) and $T_{A}=465+0.71T_{m}$ (for borates).

**Figure 4 materials-16-01127-f004:**
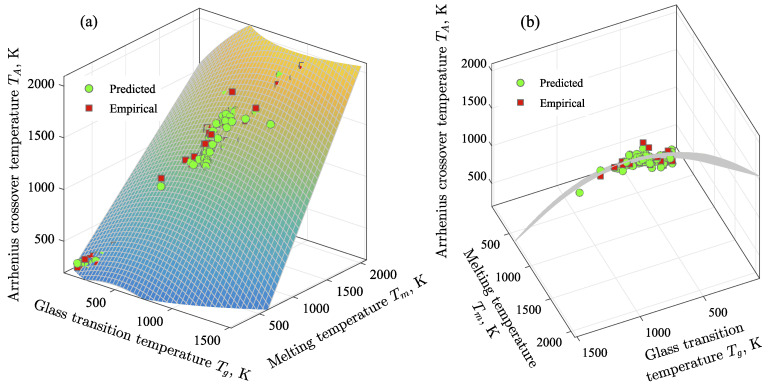
(**a**) Correspondence between the Arrhenius crossover temperature ($T_{A}$), the melting temperature ($T_{m}$) and the glass transition temperature ($T_{g}$). Circle and square markers denote predicted and empirical data, respectively. These data are compared with the results of Equation (Equation 11), which are presented as a curved surface. (**b**) This figure is from an another foreshortening, which allows one to consider the curved surface.

**Figure 5 materials-16-01127-f005:**
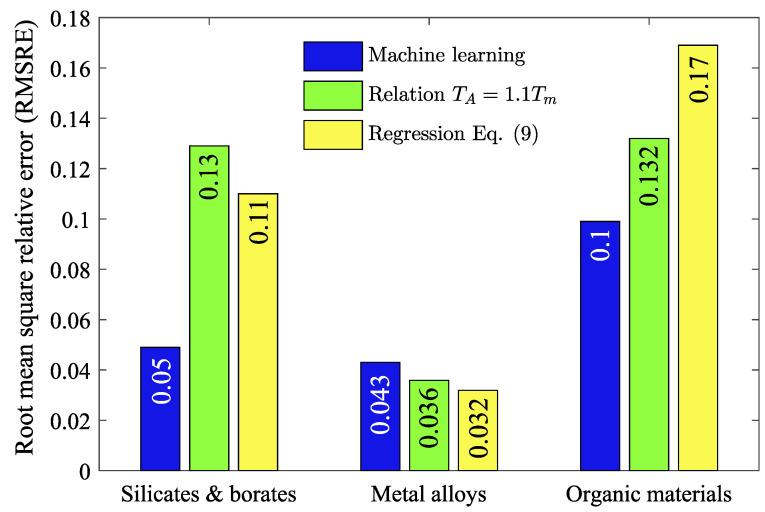
Root mean square relative error between the empirical values of $T_{A}$ and actual $T_{A}$, which is computed by different methods for silicates, borates, metallic systems and organic materials.

## Data Availability

The data presented in this study are available on request from the corresponding author.

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
