# Peer review of "Arrhenius Crossover Temperature of Glass-Forming Liquids Predicted by an Artificial Neural Network"

_materials, 2023, doi:10.3390/ma16031127_

Round 1

Reviewer 1 Report

Using a machine learning model, the authors established the relationship between the Arrhenius crossover temperature TA and the physical properties of liquids directly related to their glass-forming ability. The model developed was used to analyse silicates, borates, organic compounds and metal melts of various compositions. The work is satisfying. Thus, I believe this paper can be accepted in its current form.

Reviewer 2 Report

The manuscript by Galimzyanov et al. is a study of the Arrhenius crossover temperature TA for glass-forming liquids. Particularly, the focus is on the prediction of TA for strong glass-formers (like silicates) for which estimating TA is challenging. The work aims to predict TA from known physical features of a broad range of glass-formers (the glass transition temperature Tg, the melting temperature Tm, and the fragility index m), using a neural network model to unveil non-trivial correlations.  By minimizing the error in the prediction of TA compared to known experimental values, the authors find a non-linear equation for TA as a function of Tg and Tm, with empirical parameters obtained from a fitting procedure.

I agree that a correct theoretical determination of TA for (strong) glass-formers is a relevant research question, and that machine learning methods are important tools for this and other problems. Unfortunately, I failed to see how the results presented are an advancement compared to the current literature and simpler correlations (e.g., a linear correlation with Tm). I am not against the publication of negative results, but I cannot recommend this paper for publication unless they are properly analyzed and discussed.

In detail, several points in the manuscript make me wonder what is the improvement of the proposed model compared to a linear correlation with Tm (including the concluding remark from the authors that “the correct relationships […] are agreed with TA~1.1Tm”):

1) Figure 2a shows the error from the neural network when using different combinations of physical features as inputs. It stands out that Tm alone is almost as good as the combination of all 4 features (10.9 vs 10.5).  Unless there is an uncertainty quantification on these values and some theoretical reason, it is hard to argue that they are very different. Tm alone then could be the parameter needed to estimate TA, without the need for the neural network and of more free parameters. This is confirmed by the authors themselves in Figure 3b, where a linear correlation is shown between Tm and their predicted TA (silicates included) of TA=1.1Tm. If one estimates TA for silicates from a linear fit of the TA-Tm plot with the known Tm values, how different would it be from the TA estimated with the neural network? And how would one prove which one is more “correct”, since we have no empirical data for most of those points?

2) In figure 2b, the error for Tm alone (10.9) is lower than for Tg,Tm,m combined (11.0). How is this possible? It also points again to the fact that Tm alone seems to be the dominant parameter, since the “best” error of 10.5 is then obtained by adding again Tg/Tm (compared to the Tg,Tm,m combination), despite it being deemed “insignificant”.   

3) In equations 8-9, what is the physical meaning of the quadratic terms? And how large is the difference between this equation and TA=1.1Tm derived in Figure 3b? If the accuracy does not differ significantly, a linear equation in Tm would be preferable to a quadratic equation with multiple terms and free parameters with no physical basis.

4) As a methodology question: my understanding is that the authors used both the training and the validation set to optimize the neural network. These sets contain all systems for which TA is empirically known. I am not surprised then that the model is able to reproduce these data. Since TA for the systems in the test set is unknown, how can the reliability of the neural network results be assessed and validated? What properties are correctly predicted that were not used to calibrate the model? 

Reviewer 3 Report

Manuscript I.D.: Materials – 2139429

Arrhenius cross-over temperature of glass-forming liquids predicted by artificial neural network

Mokshin A.V., Doronina M.A., Galimzyanov B.N.

The authors propose Equation (9) to represent the Arrhenius crossover temperature predicted by an artificial neural network as a function of Tg and Tm.

It is easy for a reviewer to use this equation and to apply it to similar predictions of Tournier and Ojovan in Physica B 602 (2021) 4123542. DOI: 10.1016/J.physb.2020.412542

There, the reduced transition temperatures qn+ = qA are accompanied by exothermic or endothermic heats equal to qn+ Hm. Among these predictions, several of them were observed as shown in a parent special issue of Materials:

Tournier R.F. and Ojovan M.I. Materials, 2021,14, 2287

https://doi.org/10.3390/ma14092287

Please look at Figure 9 in Physica B where qn+ = qA is plotted versus qg with qn+ = -0.3874 qg.

Here too, qn+ depends on Tm and Tg. I recommend representing qA as a function of qg in your paper.

Now, I use Table 1 and represent Tn+ and TA versus Tm for 50 materials and finds the following straight lines:

Tn+ =1.1464 Tm and TA = 1.1229 Tm

This result (Tn+ =1.1464 Tm) agrees with your equation (TA = 1.1229 Tm) which works. Congratulations.

Round 2

Reviewer 2 Report

I thank the authors for the additional and extensive discussion of their results. I still have some doubts about the relevance and predictive power of the newly found correlation compared to the previous empirical relation between TA and Tm, but I am satisfied with the detailed discussion on the topic and I recommend the paper for publication.